# The Effects of Play Streets on Social and Community Connectedness in Rural Communities

**DOI:** 10.3390/ijerph18199976

**Published:** 2021-09-23

**Authors:** M. Renée Umstattd Meyer, Tyler Prochnow, Andrew C. Pickett, Cynthia K. Perry, Christina N. Bridges Hamilton, Christiaan G. Abildso, Keshia M. Pollack Porter

**Affiliations:** 1Department of Public Health, Baylor University Robbins College of Health and Human Sciences, Waco, TX 78628, USA; tprochnow@tamu.edu (T.P.); cbridgeshamilton@brockport.edu (C.N.B.H.); 2Department of Health & Kinesiology, Texas A&M University, College Station, TX 77843, USA; 3Division of Kinesiology & Sport Management, School of Education Research Center, University of South Dakota, Vermillion, SD 57069, USA; Drew.Pickett@usd.edu; 4School of Nursing, Oregon Health & Science University, Portland, OR 97239, USA; perryci@ohsu.edu; 5Department of Public Health & Health Education, SUNY Brockport, Brockport, NY 14420, USA; 6Department of Social and Behavioral Sciences, West Virginia University School of Public Health, Morgantown, WV 25606, USA; cgabildso@hsc.wvu.edu; 7Department of Health Policy and Management, Johns Hopkins Bloomberg School of Public Health, Baltimore, MD 21205, USA; kpollac1@jhu.edu

**Keywords:** social environment, community-based interventions, physical activity, active play, community connectedness, health externalities, social determinants of health

## Abstract

Promoting physical activity (PA) is a long-standing public health initiative to improve overall health and wellbeing. Innovative strategies such as Play Streets, temporary activation of public spaces to provide safe places for active play, are being adopted in urban and rural communities to increase PA among children. As part of these strategies, aspects of social and community connectedness may be strengthened. This study analyzes focus groups and interviews from rural Play Street implementation team members (*n* = 14) as well as adults (*n* = 7) and children (*n* = 25) who attended Play Streets hosted in rural North Carolina, Maryland, Oklahoma, and Texas to better understand the added benefits of Play Streets in community connectedness. Overall, elements of social support and social cohesion are mentioned most frequently with instrumental and conditional support; however, concepts of social capital, collective-efficacy, and social identification are also presented. Participants expressed that Play Streets provided more than just PA; they provided opportunities to access and share resources, build perceptions of safety and trust in the community, and develop relationships with others. Fostering community connection through Play Streets may reduce health inequities in rural communities by building community resilience. Community-based PA programming that enhance and capitalize on community connectedness could be effective ways to improving the overall health and wellbeing of residents.

## 1. Introduction

Physical activity (PA) and active play are lauded for their numerous physical, social, and mental health benefits for children [1,2,3,4]. Active play encompasses many concepts but can be conceptualized as creative activity in which children are physically active and is often unstructured [1]. Unfortunately, a low prevalence of children in the U.S. meet PA recommendations [5]. Children are recommended to be physically active for a minimum of 60 minutes each day for health benefits [6]. Further, research suggests that children may play outside less than previous generations [7]. Specifically, in rural communities, children face barriers to participating in PA and active play, namely few safe PA resources, decreased walkability or active transportation opportunities, long distances to resources, and concerns for safety [8,9]. In response to these barriers, rural communities often employ creative solutions to provide support for PA and PA resources [9,10].

One such creative solution is Play Streets. Play Streets is a place-based intervention in which community members temporarily (2–4 h) close a street or activate a public space, such as a field or a parking lot, to provide a safe place for children to play. While Play Streets have been used in urban areas for decades [11,12,13], rural communities have recently adapted them to fit the needs of their communities [14,15]. Play Streets typically occur episodically during the summer (June–September) to provide safe active play opportunities in a time of reduced PA and structured opportunities for activity [16,17]. Play Streets have successfully encouraged active play and PA for children in both rural and urban settings [11,14]. Reported previously, as a primary aim of this larger study, Play Streets are effective ways to promote PA and active play among children and adolescents [14] by getting people outdoors playing and interacting with others in their community.

### 1.1. Social and Community Connectedness

“Sense of community” encompasses the general idea of many interrelated concepts, including sense of community, social identification, social cohesion, social capital, collective-efficacy, social networks, and social support [18,19]. When considering their definitions and the four accepted elements that define sense of community as established by McMillan [20] of membership, influence, integration and fulfillment of needs, and shared emotional connection, we chose to consider and group these concepts under a less concrete or established frame and refer to this collective of concepts as “community connectedness”. Community connectedness includes the following interrelated concepts seen in the literature to directly or indirectly influence health: collective-efficacy, social capital (bonding, bridging, linking), social cohesion (structural, cognitive, relational), social identification (cognitive, affective, in-group ties), and social support (structural, emotional, instrumental, informational, appraisal, conditional) [19,21,22,23,24,25]. See Table 1 for definitions of each concept.

### 1.2. Benefits of Social Community Connectedness

#### 1.2.1. Health Benefits

Health benefits of concepts of social community connectedness have been well documented. Evidence supports short and long-term direct and indirect health benefits including individual health benefits across diverse populations for mental health, physical health, mortality, quality of life, stress, anxiety, sleep, chronic disease prevention and management, illness management and recovery, health-enhancing behaviors, and reduction in health disparities, with several of these also being indirectly influenced through prevention or mitigation of stress responses, access to health enhancing resources, or academic success [18,26,27,28,29,30,31,32,33,34,35]. Community-level health benefits have also been established, including direct community benefits of social well-being and empowerment, to more indirect benefits like increased community engagement, participation, and development and civic engagement [36].

#### 1.2.2. Reciprocal Relationship between PA and Community Connectedness

Social cohesion and the social environment at large are impactful for child PA [37]. Primarily, children are influenced by their social connections through peer or parental influence and social support as well as the presence of social norms [24,38,39]. For example, children are more likely to be physically active in the presence of a friend than when alone and children at rural and urban Play Streets were more likely to be physically active when there were other active children in the area [38,40,41]. Further, social cohesion and sense of community may also provide community members access to resources (i.e., social capital) which would not be available to them in a less connected community [42,43,44]. These linkages can also be helpful in the dissemination of ideas across the community such as the participation in programs like Play Streets [45].

#### 1.2.3. Community Resilience

Community resilience is a community’s capacity to adapt and recover from adversity, challenges, or natural disasters [46,47,48]. Key social factors that support community resilience are social capital, social support, collective-efficacy, and social cohesion [46,47,48]. Two years after communities experienced a natural disaster, residents in communities with high social cohesion reported feeling more prepared for another natural disaster [49], suggesting that social cohesion is an important component in disaster preparedness and community resilience. Enhancing social capital and social connectedness are approaches recommended by the U.S. government to improve community resilience [48]. Thus, an intervention that enhances these social factors has the potential to enhance community resilience.

### 1.3. Health Externalities

Health behavior interventions are designed generally with a specific purpose or outcome in mind; for example, one of the primary aims of present-day Play Streets is to promote PA among children and adolescents [12,13]. However, interventions often have unintended side effects or impacts beyond the stated purpose of the program or aim of an evaluation or study, which can be regarded as externalities. Originated in the field of health economics and also applied in other disciplines including behavioral science, externalities represent unsolicited health-related costs or benefits derived from policy, programs, or other action [50]. For example, a population reaching herd immunity (i.e., reduced risk of spread) against a particular virus due to widespread vaccination benefits not only those who get vaccinated, but also those who do not get vaccinated. An important defining feature of externalities is that they involve no additional input of resources. That is, externalities are non-primary effects of a behavior as it exists. Such second-order effects have occasionally been explored in health contexts, such as population health benefits of creating a bike-sharing infrastructure in a community [51], increased obesity stigma related to the introduction of graphic warning labels on sugary drinks [52], and water and air quality effects in neighborhoods surrounding livestock production facilities [53]. Studies have further examined externalities related to social conditions in communities, suggesting that higher social cohesion can have positive overall impacts on wellbeing for individuals and the community as a whole [54].

Existing literature suggests that small- to medium-scale shared outdoor spaces induce externalities, including social community connectedness characteristics such as enhancing social interactions and creating an atmosphere that provides an opportunity for residents to have social encounters with others [55,56]. When public spaces are inclusive, equitable, and enjoyable to inhabit, the use of the existing infrastructure increases, having the potential to improve overall quality of life in addition to PA opportunities [57]. Prior research suggests that Play Streets may foster a sense of social cohesion and sense of community within the neighborhood and can help create and strengthen a community’s social environment; however, this has not been explicitly examined [58]. Therefore, it is worth considering externalities related to health behavior interventions (i.e., health externalities) such as Play Streets that utilize (“activate”) otherwise inactive public spaces.

There are many social constructs related to social community connectedness and the benefits received through these connections. Building from these social constructs, this study aimed to understand the underlying social aspects and benefits rural communities realize when implementing Play Streets as potential health externalities. Further, this study uses multiple viewpoints from children, adults, and implementers to assess these constructs. This study focuses on five main constructs of social and community connectedness as it relates to Play Streets: social cohesion, social identification, social capital, collective-efficacy, and social support.

## 2. Materials and Methods

### 2.1. Play Street Implementation

This article comes from a larger study assessing the implementation of Play Streets in rural communities. More information on the formative procedures as well as PA outcomes can be found elsewhere [14,15]. To summarize briefly, during 2017 four partner organizations located in diverse low-income, rural communities (rural-urban commuting area (RUCA) code greater than or equal to 4.0) [59] across the U.S. were recruited based on experience implementing community events and willingness to implement Play Streets. RUCA codes classify U.S. census tracts, counties, and zip codes along the rural-urban continuum using measures of population density, urbanization, and daily commuting using U.S. census and American Community Survey data. The classification includes whole numbers 1–10 that designate metropolitan, micropolitan, small town, and rural using population size and primary commuting flows, with further detail provided on secondary commuting flows indicated by decimals. RUCA classifications define codes of 4.0–6.0 as micropolitan (population: 10,000 to 49,999), 7.0–9.0 as small towns (population: 2500 to 9999), and 10.0–10.3 as rural [59]. Given the varying definitions and diversity across rurality, a minimum RUCA code of ≥4.0 was used as inclusion criteria for recruitment of partner organizations [9]. In addition to meeting this criteria, the four participating communities were selected to represent varying degrees of rurality that were low-income, with large populations of African American, American Indian, Latino, or White, non-Hispanic residents. The four partnering communities were located in the following areas: Maryland (MD; town population = 2008; county RUCA = 10.3; 96.3% of children White, non-Hispanic), North Carolina (NC; town population = 1040; county RUCA = 10.2; 65.8% of children African American), Oklahoma (OK; town population = 1006; county RUCA = 9.0; 58% of children American Indian), and Texas (TX; town population = 5457; county RUCA = 7.1; 47.9% of children Latino) [60,61].

Each partnering community organization received a mini-grant of $6000 to implement four, three-hour Play Streets throughout the summer (June–September 2017) for a total of 16 Play Streets for the entire study. Community organizers could determine how, when, and where the Play Streets were implemented to best fit their community; however, they were required to focus on school-aged children, be open to the public at no cost, and spend at least $1000 of their grant on reusable materials or equipment such as hula hoops, frisbees, and balls. Considering community needs, each Play Street looked slightly different and most of them took place in open fields, school yards, and existing parks, as streets were not always feasible for the communities. For more information about Play Streets including pictures, guides, and descriptions please see the *Guide to Implementing Play Streets in Rural Communities* [62].

### 2.2. Recruitment

Eligible participants included Play Street implementation team members (coded ITM below), adults who attended at least one Play Street, and children (grades pre-K-6th who attended at least one Play Street). Lead implementers from each Play Street host-community recruited participants using flyers and emails disseminated to Play Streets participants and posted in publicly viewable places (e.g., elementary school doors, FaceBook page, community email distribution lists). Lead implementers invited all implementation team members to participate and contacted families with children and/or adults who attended at least one Play Street during the season with an invitation to participate.

### 2.3. Data Collection

Interviews and focus groups were conducted to document implementation. Implementation team interviews were scheduled immediately following each Play Street for formative evaluation purposes and iterative assessment. Semi-structured individual interviews and focus groups with adult and child participants also occurred 1–2 months following the final Play Street within each community to further examine implementation and PA outcomes. Post Play Streets data collection occurred during the fall following the 2017 Play Streets summer implementation as focus groups or individual interviews, depending on community partner and participants’ availability. Interviews and focus groups were conducted in person, recorded, and transcribed verbatim prior to coding.

Interview and focus group guides were similarly developed to examine implementation and outcomes of Play Streets from the three distinct perspectives of implementation team members, children, and adults. Consent, assent, and parental permission forms were signed prior to data collection. Each implementation team member and adult who participated in data collection received a $10 gift card and each child received a red 10 diameter play ball in appreciation of their time. All study methods were approved by associated institutional review boards. Child and adult participants were recruited via flyers and word of mouth by community partners; all implementation team members were invited to participate.

### 2.4. Analytic Approach

#### 2.4.1. Coding Frame

The coding frame was developed through a scoping review of the literature on social environment in health behavior. The coding frame categories, subcategories, and definitions are presented in Table 1.

#### 2.4.2. Coding Process

Four researchers were involved in the coding process. First, each researcher independently coded one transcript from an implementation team focus group. Differences were reconciled through iterative discussion and examination of the framework definitions over four meetings and the coding frame was refined. Next, each researcher coded half of two additional transcripts, one from a child focus group and one from an adult focus group. Researchers then reconciled differences in coding on these transcripts until consensus was reached for the coding frame. Next, each researcher was randomly assigned three transcripts to code (one of each type; implementation team, child, and adult). When coding of a transcript was complete, a second researcher was assigned to read the transcript and coding to check that the coding represented the meaning of the words and was in alignment with the coding frame. The two researchers then met to reconcile any differences. Since the concepts are interrelated, in some cases a piece of text could have been coded with more than one code as the meaning in the text described elements of two interrelated concepts or codes. Using four researchers with different perspectives and having a second researcher review the coding reduced bias in interpretation of the text.

#### 2.4.3. Analysis

Data were entered into NVivo, Release 1.5.1 (QSR International (Americas) Inc., Burlington, MA, USA), for analysis [64,65,66]. Since the interviews and focus groups covered similar domains, data were combined and analyzed together. Qualitative content analysis was used to analyze the data. Qualitative content analysis uses a systematic process to describe the content or meaning of the data by assigning text to categories that represent the meaning of the words in the text [64,65]. Dominant themes were identified and are presented below.

## 3. Results

Ten focus groups with a total of 43 participants (mean 4.8) and three one-on-one interviews were completed and then coded (see Table 2 for descriptive data from the focus groups and interviews), each participant attended at least one Play Street. Overall, elements of social support and social cohesion were mentioned most frequently; however, concepts of social capital, collective-efficacy, and social identification were also present. To simplify reporting, each concept is presented separately below.

### 3.1. Social Support

Social support was the most frequently identified community connectedness concept throughout data collection, regardless of participant classification (implementation team member, adult, or child). Instrumental and conditional support were the most commonly discussed. Participants also discussed informational and appraisal support.

#### 3.1.1. Instrumental Support

The most frequently discussed community connectedness concept throughout all focus groups and interviews was instrumental social support (i.e., providing a tangible service). Themes that related to this form of support noted that Play Streets provided a fun and enjoyable experience of PA for children and adults. Play Streets were also described as a safe place “I actually felt safe that we could allow our kids to go play and not have to stress on, ‘Where’s our kids? Who has our kids?’” (OK Adult), an alternative to screen activities for children in the summer, which are lacking, and a place where children have autonomy in their choices “it did give students an opportunity to have three solid hours of kind of choice” (MD ITM). Play Streets were also recognized across communities as a way to address equity challenges in access for families, especially those who are under-resourced. The lack of affordable options was acknowledged across all communities by adults, implementation team members, and children alike. Participants described this lack of options as follows:


*“And plus, it’d kind of be fun for a lot of other kids to if they don’t have anything to do, like, and if they don’t have any money to do anything. Like not go to the swimming pool or go in the pool or nothing. They can go to that and have a great time and eat and drink something for free.”*
(OK Child)


*“we don’t really have parks here and we do but it was at the recreation and not many kids have a way to get there. And it’s not—and it’s a small playground. It’s mostly for, like, infants and toddlers. There’s nothing for, like, eight and up. So yeah, it’s [Play Streets] a big difference.”*
(NC Adult)


*“There are several of our students, they don’t get a family vacation, they don’t get to go experience things that cost money so they sit at home during the summer so for them to be involved in a camp that has this as a piece of it is important to these kids, some of them, this was like all they had to look forward to in their summertime.”*
(MD ITM)

#### 3.1.2. Conditional Support 

While less frequent than instrumental support, conditional support was the second most recurring theme discussed across all four communities by children and adults, capturing the value of Play Streets in terms of opportunities for co-participation in active play and supervision. While conditional support was usually described as a benefit to children through playing with others—“We had a chance to all spend time together… it was like lovely to me when we spend time and had fun,” (NC Child) there was also evidence of benefits for adults through interacting with other adults and children—“I actually played with the kids some, which I don’t do as much as I should, so I enjoyed spending time with them and just playing and being active with them” (TX Adult). These interactions also enhanced family dynamics. One adult said that, “it [Play Streets] helps you get to know your children better, and I think because I played they trust me more and we’re able to talk and have better communication between me and the children. It’s great.” (TX Adult)

#### 3.1.3. Appraisal Support

Appraisal support, while not mentioned often, was discussed by participants from three of the four communities and most often by implementation team members or children. Implementation team members described children at Play Streets making self-evaluations, through activity goal setting, creating plans for how to get more steps (steps were tracked at the Play Streets using pedometers), or trying new physical activities while at the Play Streets. Children described appraisal support regarding self or interpersonal challenges in step counts; although one girl talked about Play Streets increasing her desire to try new sports and activities after being at Play Streets. A powerful observation was made by an implementation team member with the recognition that children experienced fun and increased confidence through PA at Play Streets, “I don’t even think they realized they were exercising or doing something that involved exercise. All they knew is they were having fun or jump roping... once they succeeded that it was like wow, I can do this. So, I think my experience was listening and being there I think it helped a lot of kids realize that there’s other things to do out there besides video games.” (OK ITM)

#### 3.1.4. Emotional Support

Emotional support was directly discussed, versus implied, by children and an adult, from two of the communities. Children described Play Streets making them feel “happy” (NC Child, TX Child). The adult described Play Streets bringing laughter and enjoyment of company with others.

#### 3.1.5. Informational Support 

Interviewees rarely discussed informational support; however, a child and implementation team member, both from MD, acknowledged the inclusion of running technique and health education components at Play Streets, respectively. 

#### 3.1.6. Structural Support

Structural support was also rarely discussed directly but included recognition by an adult that Play Streets provided an opportunity for increased positive or healthy social network connections for children, versus the alternative of “going to get with the wrong crowd” (OK, Adult). One child also described having increased familial structural support for interactions with her brother, “I got to play with my brother a lot. And he usually wouldn’t let me play with him at home that much” (TX Child). 

### 3.2. Social Cohesion

Within the social cohesion subcodes, structural social cohesion was mentioned by most of the participants. In this manner, individuals were more often discussing that Play Streets provided them the opportunity to socialize or connect with others. One child said, “We had a chance to all spend time together... It was like lovely to me when we spend time and had fun” (NC Child). An adult stated, “I know that they always need something out here, and it’s really nice to have out here and it brings that community together” (OK Adult). Another adult mentioned, “For us, well, we’ve only been here two years, and I feel like the community is kind of closed-off. There’s the groups of people who’ve known each other forever, and I think Play Streets helped me get to know some of the people better and I guess make better friends” (TX Adult). 

Additionally, adults brought up elements of social cohesion more than the other two participant types. Within these comments they also mentioned concepts of cognitive social cohesion or feelings of trust or shared values through attending Play Streets which often manifested feelings of safety. One adult put it this way, “But even if something like this was to happen, and say the parents could not make it, at least we would know our kids were safe having fun” (OK Adult). One adult also stated, “I felt safe, felt that my kids were safe, and I didn’t have to keep as close an eye on them as I do most of the time” (TX Adult).

### 3.3. Social Capital

Play Street’s impact on community social capital was also mentioned by adults and implementation team members from all four communities, particularly in the forms of bridging and linking capital. Generally, discussions of social capital in our data took the form of individuals discussing the increased access to social networks and resources provided by engagement with Play Streets. For example, several participants noted that Play Streets afforded children the opportunity to engage in activities to which they otherwise may not have access. Interestingly, the two forms of social capital identified in our data were most discussed by different interviewee types (i.e., adult participants and implementers), reflecting the respective interests of each. 

#### 3.3.1. Bridging

Bridging social capital, whereby resources can be accessed across different socioeconomic or demographic groups, was most often discussed among adult participants. Generally, this related to Play Streets giving children access to forms of activity they would not otherwise engage. One participant, for example, said, “I’d seen some kids that really enjoyed some of those things that probably never would have gotten the opportunity to go somewhere and do something like this to play on bounce houses and things like that; just different income levels and those sort of things” (OK ITM). 

#### 3.3.2. Linking

Among the implementation team members, linking social capital was more commonly mentioned. Linking social capital is defined by the creation and maintenance of relationships with formalized institutions or structures. Implementers seemed to be conscious of the opportunity inherent in Play Streets to connect participants with various forms of support available to them, intentionally building in elements involving local health departments, libraries, law enforcement, and religious institutions into the intervention. In some cases, Play Streets were held in conjunction with other support programs or community events to maximize feasibility, reach, and overall impact. For example, one Play Street was held in conjunction with a local back-to-school backpack drive. Thus, attendees were able to play and be active at the Play Street while also receiving school supplies for the upcoming year. For implementers, this type of strategic connection, wherein Play Streets locations were chosen to introduce and maximize access of community members to institutional resources (i.e., linking social capital), was important.

#### 3.3.3. Bonding

While we included bonding social capital as part of our coding frame, it was not described explicitly or through a related theme by any participants. 

### 3.4. Collective-Efficacy

The belief in the interviewees’ community’s ability to work together and successfully implement expanded and improved Play Streets the following year was touched upon by three communities. In two of the Implementation Team focus groups, implementers talked about other organizations or key stakeholders they could engage to enhance the Play Streets. They also described community members, including children helping run the event and “playing a part.” One of the implementers explained, “But that’s how we work, you know? When we’re committed to something, we’re going to make it work. And that’s what we did.” (NC ITM) A few participants in one of the adult focus groups described being able to organize and provide transportation to bring families from other communities together for the Play Streets. As one of the adults stated, “I think frankly something like that [providing transportation] was presented to us, there would be volunteers to even—like say they knew somebody that could not make it.‘Hey, I got the extra room. Would you like to go?’” (OK Adult)

### 3.5. Social Identification

Only a few participants across the three communities described themes related to social identification. In two of the adult focus groups, participants described bonding with other parents at Play Streets. They described being able to interact with other parents or adults while their children were safe and playing. They enjoyed “being able to communicate with other adults.” (OK Adult). One of the implementers explained that a benefit of Play Streets was that the children who participated developed a sense of belonging. “They became like a community on—they became known as like the Play Streets kids and they liked that, they liked being part of that group.” (MD ITM)

## 4. Discussion

This study contributes to a key gap in the literature by exploring health externalities present when implementing four Play Streets in rural communities, specifically those related to social connectedness. Throughout this analysis, core elements of social connectedness were expressed from multiple viewpoints (i.e., children, adults, Play Streets implementers). Participants in the focus groups and interviews expressed that Play Streets provided more than just PA; they provided opportunities to access and share resources, build perceptions of safety and trust in the community, and develop relationships with others. 

### 4.1. Access to Community Resources

Barriers to PA, including safety and a lack of safe places to be active, are inequitably distributed and often disproportionately impact some communities, including rural communities, communities of color, and low-income individuals [8,9,67]. Participants described how Play Streets can help address equity challenges within rural communities, especially among under-resourced families who may not be able to afford summer opportunities that require fees for participation (i.e., camps, swimming pools, organized sports, etc.). This is especially relevant in summer when PA levels have been shown to decrease [68]. By providing summer opportunities for play and connection that are free and accessible to many children and families, communities can address barriers to health equity such as cost and access.

Beyond access to PA, Play Streets were also described as providing community members access to community health resources they may otherwise not have encountered. Implementation teams were intentional in efforts to pair Play Streets with other community resources to maximize benefits for participants. For example, one Play Street was held in conjunction with a local back-to-school backpack drive, providing children from low SES backgrounds with school supplies for the upcoming academic year, with other partners including WIC, the health department, the public library, and a summer meals program. Given their highly visible nature (e.g., inflatables, tents, large gatherings), Play Streets often attract both planned attendees and the passerby. By including local community organizations, Play Streets provided access to various community resources in a location with multiple benefits and increased awareness of potentially unused resources. In rural settings, where shutting down a major thoroughfare is impractical, Play Streets were often implemented in proximity of community centers, public libraries, or faith-based institutions. Given the value of such programs in addressing disparities, Play Streets served as a tool to promote equity in communities. 

### 4.2. Community Safety and Perceptions of Trust

Safety or the perception of safety in the community at Play Streets was one of the social benefits which came through strongly in several focus groups and interviews. Participants noted they felt safe themselves or they felt safe allowing their children to play at Play Streets. In a related study, perception of safety and particularly traffic concerns have been negatively associated with daily PA in rural adults [69]. Similarly, lack of perceived safety is noted as a major deterrent for rural parents to allow their children to play outside [70]. Play Streets could provide opportunities for the community to create connections, forming social cohesion, and in turn fostering a greater perception of safety among its members to promote positive health behaviors like active play and PA.

### 4.3. Developing Social Relationships

Having an opportunity to create connections among children and adults was mentioned by many of the participants. This opportunity to bond or create connections, which may not have happened in the absence of Play Streets was labeled as structural social cohesion in the present framework, but it can also be the foundation on which other social constructs are built. Connections-forming can be the groundwork for other social benefits that were described by the participants [18,26,27,28,29,30,31,32,33,34,35].

### 4.4. Implications

Given the interplay between social connectedness and health, this research suggests a need for more PA interventions that are intentionally created to develop connected, cohesive, and supportive communities. While Play Streets in this study were implemented with the goal of increasing safe opportunities for active play and PA for children, there were also health externalities in the form of social impacts. Creating enhanced community connection through Play Streets may be an avenue to reduce health disparities in rural communities while also building community resilience. Enhancing community connections creates opportunities for additional health benefits above and beyond those from PA, suggesting that PA and active play could be a multi-purposed vehicle to community-level health and wellbeing. It is important to note that while not included in the present study, these spaces need to be inclusive to help realize benefits related to social connections. 

Play Streets and similar community-based PA interventions also provide opportunities for collaboration with other health partners and initiatives. Communities may consider including or “coupling” Play Streets with other health promotion initiatives or community services (e.g., health fairs, nutrition seminars, vaccination clinics, safety demonstrations), allowing the community to address multiple health objectives simultaneously while also addressing a potential barrier of transportation and access [15]. Given disparities facing rural populations, providing multiple health resources as part of a single community-event may be particularly effective and increase accessibility. This judicious approach also has potential reciprocal benefits for a community by increasing access to and opportunities for PA and active play when it is part of other community efforts and Play Streets also serving as a means to draw engagement with other community and health resources. It should be noted that existing community resources and organizations may influence the types of organizations and individuals most “ready” to successfully implement Play Streets and similar initiatives. Thus, implementers should pursue partnerships most appropriate for their specific communities. Play Streets can be initiated and implemented by a variety of partners and can include anyone seeking to promote health and/or foster relationships within their communities (e.g., advocates, community-based organizations, faith-based institutions, health departments, hospitals (including residents), libraries, neighborhood associations, policy makers, public servants, schools) [62].

Finally, while the current study focused on the benefits of implementing multiple Play Streets over a limited time period (i.e., one summer). These findings and evaluations out of Chicago and other urban areas suggest value in scheduling Play Streets that recur across multiple years and also recur multiple times within a year [11,12,13]. Important considerations for implementers include logistical challenges (e.g., location, date/time, activities, bathrooms), the human resources needed, identification of active play equipment and partners, marketing, and retaining free access to all families, and the potential of competing demands and/or loss of novelty across time. However, our findings suggest a strong potential for enhancing social connectedness characteristics from more regular implementation of Play Streets in rural communities. Because of the benefits of Play Streets, rural communities should consider how to sustain them. Local policies could be implemented to facilitate the approval process to close streets to traffic, when feasible, or that support joint/shared use of school and community facilities and places to locate Play Streets [71].

### 4.5. Limitations

While this study identifies multiple social connectedness benefits as health externalities of Play Streets, a few limitations need to be considered. Most notably, the interview and focus group guides for this study were designed to examine implementation and PA outcomes, without direct consideration of the social connectedness framework presented in this paper. Because of this we did not ask questions framed specifically for these concepts and coding was impacted, specifically considering the inter-connectedness of the codes which creates a challenge for discriminatory validity. However, the identification of these health externalities also provides strength to the study in that these conceptual themes emerged unprompted. Given the salience of these concepts in the current study, future researchers should consider a more direct, a priori examination of these concepts and how Play Streets and other community-based PA interventions can influence community connectedness. Additionally, considering the study design, there are possible limitations given the potential response bias, not knowing if participants’ responses were due to the intervention itself or the perception of the intervention, and the lack of pre-intervention data for comparison. Certain limitations are also possible in focus group research (e.g., respondent fatigue, group dynamics), however we have sought to minimize such effects in the current study by employing best practices related to group size and number, homogeneity, and topical clarity [72,73]. Additional research may examine dynamics of social connectedness in PA interventions using different methods (e.g., survey) to further develop understanding. Future research should also include additional rural communities. While the four communities included in this study represent multiple layers of diversity seen throughout rural America, they do not represent all rural communities.

## 5. Conclusions

As noted, the primary aim of this Play Streets project was to increase active play opportunities for and PA of children. However, the health externalities seen through enhanced social opportunities provided throughout these communities were evident when speaking with adult and child participants as well as the implementation teams. Play Streets provided opportunities to create connections and strengthen existing ones, which in turn provided community members with far-reaching social benefits not limited to the health benefits of PA alone. Community-based PA interventions and programming (such as Play Streets) that enhance and capitalize on community connectedness are effective ways to reduce health disparities and improve the overall health and wellbeing of residents. While PA benefits are sufficient reason for community-based active play interventions like Play Streets in the short-term, the additional benefits of increasing community connectedness should be considered as an important vehicle toward improved health and wellbeing for rural communities.

## Figures and Tables

**Table 1 ijerph-18-09976-t001:** Coding frame categories, subcategories, and definitions.

Code	Definition	Citation
Social Cohesion	“The presence of strong social bonds that bridge divisions in society, and the lack of conflict in a society. The sense of solidarity among members of a community.”	[23]
Structural	“Formal opportunities in which individual actors might develop social ties or social networks.”	
Cognitive	“Perceptions of trust, reciprocity, and support, shared values.”	
Relational	“Nature of relationships and identification with others.”	
Social Identification	“An individual’s socio-cognitive identification with one or several social category/ies or with one or several concrete groups.”	[23]
Cognitive	“Self-categorization of belonging to group, or group membership.”	
Affective	“Emotional evaluation of group membership.”	
In-group ties	“Perceptions of similarity and bonds with group members.”	
Social Capital	Shared resources that individuals have access to through their social networks. The effective functioning of social groups through interpersonal relationships, a shared sense of identity, a shared understanding, shared norms, shared values, trust, cooperation, and reciprocity.	[23,25,63]
Bonding	Resources that are accessed within networks or groups having generally similar characteristics.	
Bridging	Resources that may be accessed across groups of different socioeconomic or sociodemographic characteristics.	
Linking	The norms of respect and networks of trust connecting individuals and groups across formal or institutionalized structures of authority and power.	
Collective-efficacy	It is grounded on mutual trust and describes a community’s ability to create change and exercise informal social control (i.e., influence behavior through social norms).	[19]
Social Support	A network of family, friends, neighbors, and community members that is available in times of need to give psychological, physical, and financial help.	[21,22,24]
Structural	Size of the network and frequency of social interactions.	
Emotional	Expressions of emotions, usually empathy, love, trust, and caring.	
Instrumental	Tangible aid and service.	
Informational	Advice, suggestions, and information.	
Appraisal	Information that is useful for self-evaluation.	
Conditional	Supervision or co-participation.	

**Table 2 ijerph-18-09976-t002:** Post Play Streets focus group and interview descriptive information.

	Maryland	North Carolina	Oklahoma	Texas
Imp Team	1 FG (*n* = 3)	1 FG (*n* = 2)	1 FG (*n* = 4)	1 FG (*n* = 5)
Adults	0 *	2 Interviews	1 FG (*n* = 4)	1 Interview
Children	1 FG (*n* = 9)	2 FGs (*n* = 6)	1 FG (*n* = 6)	1 FG (*n* = 4)

* The viewpoint of adults was included in the Implementation Team focus group. Imp Team = implementation team; Adults = participating adults; FG = focus group; Interview(s) = individual interview(s).

## Data Availability

The data presented in this study are available on request from the corresponding author.

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
