# Peer review of "The Effects of Play Streets on Social and Community Connectedness in Rural Communities"

_ijerph, 2021, doi:10.3390/ijerph18199976_

Round 1
Reviewer 1 Report
Will be brief in my comments due to writing limitations.
The MS addresses an interesting question. Yet, the post-hoc qualitative approach taken, presents several limitations which should be acknowledged in the discussion: a) likely suggestibility effects, b) not knowing whether the intervention itself, or the perception of the intervention (or both) impacted reactions, c) not having pre-intervention data.
Author Response
Thank you for these suggestions regarding the limitations of the findings. The suggested additional limitations have been added to the 4.5 Limitations section (final sentence of this paragraph).
Reviewer 2 Report
Dear authors
Let me add some thoughts and suggestions to the manuscript.
1. The manuscript presents a very interesting idea "Play Streets". Where I live this strategy is known as "leisure streets". It strengthens the concept of health promotion that goes far beyond strategies/actions in formal health facilities. Based on the experience here, I ask you if there would be a possibility of other health professionals working in these spaces (nurses, nutritionists, psychologists), in a sense of an integral health care to the rural population - considering their vulnerabilities and risks?
2. Also in the country where I live and work, it is not possible to offer gift cards with monetary values. Considering that this is also the reality in many other countries, how could this strategy can be adjusted?
3. It is understood that conducting many focus groups generates a burnout of the topic. Did you notice this problem among the evaluated groups? I recommend reporting it in the text.
4. A point that I also question, and that could be part of the discussion, is about the expansion of this proposal as a permanent public policy. I think it is worth working on some hypotheses about what would need to be added and removed for large scale implementation in rural contexts (and involving populations with different particularities).
Author Response
Reviewer 2 Comments:
Comments and Suggestions for Authors
Dear authors
- The manuscript presents a very interesting idea "Play Streets". Where I live this strategy is known as "leisure streets". It strengthens the concept of health promotion that goes far beyond strategies/actions in formal health facilities. Based on the experience here, I ask you if there would be a possibility of other health professionals working in these spaces (nurses, nutritionists, psychologists), in a sense of an integral health care to the rural population - considering their vulnerabilities and risks?
- We have added to the discussion to address this concern. We have suggested the inclusion of other health resources (e.g., nutritionists) at events. We also note the importance of existing community resources in finding appropriate partnerships and determining who is most able to facilitate Play Streets and similar initiatives.
- Also in the country where I live and work, it is not possible to offer gift cards with monetary values. Considering that this is also the reality in many other countries, how could this strategy can be adjusted?
- We understand that not all researchers will be able to provide financial incentives for participants. We have also previously conducted similar studies without monetary incentives. Based on our experience, the research process did not differ significantly due to the participation incentives in this study. Given the relatively small value of the gift cards ($10), we do not believe they significantly impacted the sampling or research process.
- It is understood that conducting many focus groups generates a burnout of the topic. Did you notice this problem among the evaluated groups? I recommend reporting it in the text.
- Text was added to the limitations to include potential for respondent fatigue. However, we believe this is a limited concern in this study, given each participant only sat for a single focus group session. As such, we believe topic burnout was unlikely in this case.
- A point that I also question, and that could be part of the discussion, is about the expansion of this proposal as a permanent public policy. I think it is worth working on some hypotheses about what would need to be added and removed for large scale implementation in rural contexts (and involving populations with different particularities).
- We agree that there are policies that can support expansion and permanency of Play Streets. We added text related to this comment in the Implications section (4.4). We also added discussion regarding the potential for long term/ permanent implementation in this section. We note, particularly, that there are several important concerns related to a more regular implementation, including: funding and loss of novelty effects. Therefore, we suggest future work to explore how Play Streets could be implemented more regularly in rural communities.
Reviewer 3 Report
This is a valuable contribution to the literature, and the authors should be commended for writing a succinct report of their study. However, I think that the sudy could be more closely linked to the cited barrier to outdoor play, as opposed to just calling social connectedness a health externality - these ideas are becoming crucial. I think the findings would have more impact if placed in this context. Overall a sound article, but could be better written in places. My comments for how to do so are below.
Abstract
- Please make it clearer who the participants were - parents, organisers, children
- How many participants
- Where the Play st were
Intro
Perhaps need to make clear that 1' outcome was PA, but this study was on social connectivity etc.
Line 44 - incorrect grammar. Should read ...PA and active play namely; few safe...
1.1 Health externalities - wonder if a more relevant example i.e. Related to PA could be used instead of vaccines?
The oder and flow of points is logical, but I think the focus/emphasis on social connectedness needs to come at the beginning.
Materials and Methods
2.1 - commuting area code needs to be explained as this unit is not common to all countries
2.2 Lines 177-184 are difficult to understand on first reading as each group has not been defined at this stage. Consider rephrasing and reodering as per below comments re. Recruitment. Data collection methods and timeline might be clearer as a flow diagram as it is difficult to follow what is happening from the description.
Recruitment details should come before data collection procedures - writing chronologically will help with clarity. Also more detail required on this.
2.3.1 - thank you for presenting the coding frame
2.3.2 - robust coding practice
Results
Overall well written. Clear link to coding structure
Line 213 - Change to "Ten focus groups with a total of 43 participants..."
Please provide an indication of how many people in each category attended the Play streets.
3.1.2 Conditional support - would be good to have a few more quotes to demonstrate that it was a significant theme discussed.
Line 292 - Is racked the best word here?
Discussion
Line 396 - avoid overstating, use "contributes to a key gap in the literature..."
Line 397 - 4 rural communities in the US
4.1 - should refer to literature examining barriers to PA as inequity is a big factor in this. The reference back to PA would highlight the significance of looking at social connectedness in the context of Play streets for PA as has been done for 4.2.
Author Response
Reviewer 3 Comments:
Comments and Suggestions for Authors
This is a valuable contribution to the literature, and the authors should be commended for writing a succinct report of their study. However, I think that the study could be more closely linked to the cited barrier to outdoor play, as opposed to just calling social connectedness a health externality - these ideas are becoming crucial. I think the findings would have more impact if placed in this context. Overall a sound article, but could be better written in places. My comments for how to do so are below.
Abstract
- Please make it clearer who the participants were - parents, organisers, children,
- How many participants
- Where the Play st were
- Thank you for these suggestions; all suggestions for the Abstract section were addressed.
Intro
Perhaps need to make clear that 1' outcome was PA, but this study was on social connectivity etc.
- Text was added in the introduction to clarify this.
Line 44 - incorrect grammar. Should read ...PA and active play namely; few safe...
- Upon review of the suggestion, the grammar is correct regarding the text on Line 44 (original draft), and was not edited.
1.1 Health externalities - wonder if a more relevant example i.e. Related to PA could be used instead of vaccines?
The order and flow of points is logical, but I think the focus/emphasis on social connectedness needs to come at the beginning.
- The co-authors also originally discussed the possibility of ordering the Introduction as suggested when conceptualizing the manuscript; upon review we have modified the order and flow of the introduction to reflect the reviewer’s suggestion. Thank you.
Materials and Methods
2.1 - commuting area code needs to be explained as this unit is not common to all countries
- Thank you for this reminder, an explanation has been added regarding the RUCA classification system used in this study.
2.2 Lines 177-184 are difficult to understand on first reading as each group has not been defined at this stage. Consider rephrasing and reodering as per below comments re. Recruitment. Data collection methods and timeline might be clearer as a flow diagram as it is difficult to follow what is happening from the description.
Recruitment details should come before data collection procedures - writing chronologically will help with clarity. Also more detail required on this.
- As suggested we re-orderd the Methods section to flow more chronologically. A new section titled “Recruitment” was added and moved ahead of the “data collection” methods section. Additional detail was also added regarding recruitment methods. This re-ordering of the details included in the methods section should help clarify the procedures. Thank you for these suggestions.
2.3.1 - thank you for presenting the coding frame
- You are welcome and thank you for this feedback.
2.3.2 - robust coding practice
- Thank you for this feedback.
Results
Overall well written. Clear link to coding structure
Line 213 - Change to "Ten focus groups with a total of 43 participants..."
Please provide an indication of how many people in each category attended the Play streets.
- These suggested edits were addressed. Thank you.
3.1.2 Conditional support - would be good to have a few more quotes to demonstrate that it was a significant theme discussed.
- Thank you, an additional quote from a child was added to this section.
Line 292 - Is racked the best word here?
- Thank you, this was a typo and has been corrected to read “tracked”. Again, thank you for your detailed review and for this question.
Discussion
Line 396 - avoid overstating, use "contributes to a key gap in the literature..."
- Thank you, this modification in our wording was made.
Line 397 - 4 rural communities in the US
- This change has been made.
4.1 - should refer to literature examining barriers to PA as inequity is a big factor in this. The reference back to PA would highlight the significance of looking at social connectedness in the context of Play streets for PA as has been done for 4.2.
- Thank you for this suggestion. A sentence was added to section 4.1.